# Validating the classics: Accurate reference gene panel for reliable RT-qPCR in Porifera

Kseniia V. Skorentseva[1], Nikolai P. Melnikov[2,3], Alexander V. Ereskovsky[1,4], Andrey I. Lavrov[3], Aleena A. Saidova[5,6]*

1 Laboratory of Morphogenesis Evolution, Koltzov Institute of Developmental Biology of Russian Academy of Sciences, Moscow, Russia, 2 Department of Invertebrate Zoology, Faculty of Biology, Lomonosov Moscow State University, Moscow, Russia, 3 Pertsov White Sea Biological Station, Faculty of Biology, Lomonosov Moscow State University, Moscow, Russia, 4 Aix Marseille University, Institut Méditerranéen de Biodiversité et d'Ecologie (IMBE), CNRS, IRD, Avignon University, Marseille, France, 5 Faculty of Biology, Shenzhen MSU-BIT University, Shenzhen, China, 6 Department of Cell biology and Histology, Faculty of Biology, Lomonosov Moscow State University, Moscow, Russia

* aleena.saidova@gmail.com

## Abstract

Quantitative real-time PCR (RT-qPCR) is a powerful method for gene expression analysis, but its accuracy critically depends on the selection of stable reference genes for normalization. In non-model organisms such as sponges (phylum Porifera), this task is complicated with high biological variability, seasonal fluctuations, and limited molecular resources. In this study, we first identified and validated candidate reference genes in the calcareous sponge *Leucosolenia corallorrhiza*. Seven commonly used housekeeping genes (*ACT1, GAPDH, RPL13A, HPRT1, RPS3A, TBP, LMN1*) were selected based on available transcriptomic and genomic data, and their expression stability was evaluated using geNorm, NormFinder, BestKeeper, and RefFinder under physiological conditions and during tissue regeneration. *RPL13A, ACT1*, and *GAPDH* were identified as the most stable reference genes in *L. corallorrhiza*. To assess whether reference genes identified in *L. corallorrhiza* can be applied more broadly, we extended the analysis to three phylogenetically and ecologically distinct sponge species: *Halisarca dujardinii* (marine demosponge), *Ephydatia fluviatilis* (freshwater demosponge), and *Lycopodina hypogea* (carnivorous marine demosponge). The same panel of candidate genes was evaluated in all species using the same analytical approaches. Although similar subsets of genes (including *RPL13A, ACT1,* and *GAPDH*) consistently ranked among the most stable candidates, no single gene exhibited universal stability across all species. Pairwise variation analysis indicated that the use of two reference genes is sufficient for accurate normalization, while the geometric mean of three top-ranked genes further improves reproducibility and reduces the risk of false-positive results, as demonstrated performing *RHOA* normalization. Overall, our results demonstrate that reference gene stability in sponges is species-specific and cannot be reliably predicted based on

**Data availability statement:** Raw RT-qPCR runs' data and R script used for data analysis in this study are available in the Mendeley Data repository, doi: 10.17632/stkgvs9s26.1.

**Funding:** The research was supported by the Russian Science Foundation project no. 23–74–10005 (studies in Leucosolenia corallorrhiza and Halisarca dujardinii) and no. 24-14-00452 (studies in Ephydatia fluviatilis and Lycopodina hypogea).

**Competing interests:** The authors have declared that no competing interests exist.

**Abbreviations:** *ACT1*, β-cytoplasmic actin-like gene; CR, 'crescent' regenerative membrane; CV, coefficient of variation; Efl, *Ephydatia fluviatilis*; FSW, filtered seawater; *GAPDH*, glyceraldehyde 3-posphate dehydrogenase; hpo, hours post-operation; Hdu, *Halisarca dujardinii*; *HPRT1*, hypoxanthine phosphoribosyl-transferase 1; IT, intact tissue; Lco, *Leucosolenia corallorrhiza*; Lhy, *Lycopodina hypogea*; LMN1, lamin gene; NF, normalization factor; *RHOA*, Ras homolog family member A; *RPL13A*, ribosomal protein L13a; *RPS3A*, 40S ribosomal protein S3a; RT-qPCR, quantitative real-time PCR; SD, standard deviation; *TBP*, TATA box binding protein.

ecological or phylogenetic grouping alone. At the same time, we define a robust panel of candidate reference genes that can serve as a starting point for RT-qPCR studies in Porifera, provided that species- and condition-specific validation is performed. Our study also highlights that technical challenges inherent to research on non-model organisms must be carefully considered in the design of future studies.

## Introduction

Quantitative real-time PCR (RT-qPCR) is one of the most widely used techniques for measuring gene expression due to its high sensitivity, specificity, and dynamic range. However, obtaining accurate and reproducible results depends critically on the proper normalization of expression data, which compensates for variations in an experimental setup [1–4]. This is typically achieved using internal reference genes, which are assumed to maintain stable expression across samples and conditions. Despite the availability of clear guidelines for reference gene validation [5], many studies, especially in invertebrates, rely on housekeeping genes traditionally validated in vertebrate models, without empirical testing in the species of interest. Notably, even in vertebrate systems, it has been shown that reference gene expression may vary significantly across tissues, developmental stages, and experimental conditions [6,7], thus requiring careful, context-specific validation.

Accurate normalization is especially important in sponges (phylum Porifera), which represent an early-branching metazoan lineage of high evolutionary interest and exhibit considerable biological variability, making the validation of reference genes particularly critical. Sponges possess highly plastic, regeneration-capable tissues, which undergo seasonal, environmental and physiological remodeling, and differ drastically from bilaterians in their cellular architecture and physiology [8–10]. The molecular biology of sponges also differs from other animals, as poriferans often show unique composition of many gene families/gene programs (either lacking specific orthologs or undergoing Porifera-specific paralogous expansions). Moreover, many sponge species possess a calcareous or siliceous skeleton, which introduces additional biochemical challenges to RNA extraction and may contribute to residual contaminants, such as calcium ions and polysaccharides, potentially interfering with enzymatic reactions in RT-qPCR [11]. Overall, studies on the molecular biology of sponges require proper identification of a reliable set of reference genes specific to this early branching group of metazoans.

Although gene expression studies in sponges are gaining momentum owing to their evolutionary and ecological significance [12–15], systematic validation of reference genes remains virtually absent. Previous RT-qPCR studies in invertebrates have frequently used the *ACTB*, *GAPDH*, and ribosomal protein genes, with varying degrees of empirical support [16–20]. In Porifera, however, no standardized panels of validated reference genes have been established to date [10,21], raising concerns regarding the reliability of the expression data generated in this phylum.

To address this gap, we conducted a systematic evaluation of candidate reference genes in the calcareous sponge *Leucosolenia corallorrhiza*, which serves as a model system for regeneration studies. To select candidate reference genes, we focused on commonly used housekeeping genes representing different functional classes. *ACT1* encodes actin, a major component of the cytoskeleton involved in cell structure and motility. *GAPDH* is a key glycolytic enzyme frequently used as a reference due to its role in central metabolism. *RPL13A* and *RPS3A* encode ribosomal proteins of the large and small subunits, respectively, and are associated with protein synthesis. *HPRT1* is involved in purine metabolism and is widely used as a normalization control in gene expression studies. *TBP* encodes the TATA-binding protein, a general transcription factor essential for RNA polymerase II initiation. *LMN1* encodes lamin intermediate filament proteins forming the nuclear lamina and contributing to nuclear structure and genome organization. These genes were selected to cover distinct cellular processes, reducing the likelihood of co-regulation and increasing the robustness of normalization. Their expression stability was analyzed across three biologically distinct conditions, reflecting different regeneration stages: intact tissues (IT), body fragments with a growing regenerative membrane (CR), and body fragments with a fully sealed regenerative membrane (RM). To assess reference gene stability, several statistical algorithms have been developed, each based on distinct principles. geNorm evaluates gene expression stability by calculating the average pairwise variation between candidate genes and identifies the optimal number of reference genes required for normalization [5]. NormFinder uses a model-based approach to estimate both intra- and inter-group variation, providing a direct measure of expression stability [22]. BestKeeper relies on raw cycle threshold (Ct) values to assess gene stability based on standard deviation and coefficient of variation [23]. RefFinder integrates the results of these methods to generate a comprehensive ranking of candidate genes [24]. Because each algorithm applies different assumptions and statistical criteria, their combined use is recommended to obtain a more robust and reliable evaluation of reference gene stability. For the robust assessment, we further tested the impact of reference gene selection on the quantification of the regeneration-associated gene *RHOA*.

To ensure the broader applicability of the identified reference genes and evaluate their interspecies' stability, we extended the analysis to three additional species representing class Demospongiae and diverse ecological backgrounds: *Halisarca dujardinii* (marine demosponge), *Ephydatia fluviatilis* (freshwater demosponge), and *Lycopodina hypogea* (carnivorous marine demosponge).

## Materials and methods

### Sampling

Specimens of *Leucosolenia corallorrhiza* Haeckel, 1870 (Leucosolenida, Calcarea, Porifera) [25] were collected in Kandalaksha Bay, the White Sea, at the outskirts of the Pertsov White Sea Biological Station of Lomonosov Moscow State University (66°34′ N, 33°8′ E) from June to August 2023. Animals were collected in the upper subtidal zone (0–2 m) during low tides and kept in a laboratory aquarium with natural seawater for no longer than five days.

For the regeneration assay in *L. corallorrhiza,* we distinguished three time points: intact tissues (IT); body fragments with a growing regenerative membrane (CR as in 'crescent'), 24 hours post-operation (hpo); body fragments with a fully sealed regenerative membrane (RM), 48 hpo. IT samples (*N* = 4) were retrieved by processing whole sponges and clearing debris with minimal damage. As a regeneration model, we used ring-shaped body fragments from oscular tubes. They were cut with Castroviejo scissors, collected, and cleared of debris. To obtain ring-shaped body fragments, oscular tubes were transversely cut into rings 0.5–1.5 mm in width (osculum rims were excluded). All operational procedures were performed manually under a stereomicroscope in 0.22 μm-filtered seawater (FSW). The fragments were sustained in 30 mm plastic Petri dishes with 5 ml of FSW, half of which was changed every 12 h. Body fragments were maintained at physiological temperatures for the species during the summer period (8–12°C). Regular observations under a stereomicroscope were performed to exclude contaminated samples and evaluate the stage of regeneration. Each replicate contained more

than 100 ring-shaped fragments obtained from oscular tubes of 1–3 sponges (for CR samples, $N=6$, and for RM samples, $N=4$). The differences in sample size were due to technical limitations encountered during sample processing.

Specimens of marine sponge *Halisarca dujardinii* Johnston, 1842 (intact tissues, $N=6$) were also collected from the upper subtidal zone of Kandalaksha Bay in February 2025. Animals were kept in a laboratory aquarium with natural seawater for no longer than two days.

Gemmules of the freshwater *Ephydatia fluviatilis* Linnaeus (1759) were collected manually from an unnamed small river flowing from the Vaulino Reservoir (v. Vaulino, Sergievo-Posadsky District, Moscow region) in November, 2023. The gemmules were immediately placed in cold water and stored at 2°C until further use. To start germination, gemmules were placed in sterile Mineral-medium, M-medium (1 mM $CaCl_2x6H_2O$, 0.5 mM $MgSO_4x7H_2O$, 0.5 mM $NaHCO_3$, 0.05 mM KCl, and 0.25 mM $Na_2SiO_3$) [26] at room temperature (RT, 22°C). Each IT sample ($N=5$) consisted of 40–50 young (7 days after hatching) fully developed sponges.

The culture of the marine carnivorous sponge *Lycopodina hypogea* Vacelet & Boury-Esnault (1996) is kept at 2.5–3 L containers with artificial seawater (14°C) at the Marine Station d'Endoume (Marseille). Under such conditions (weekly feeding with *Artemia salina* nauplii and water change), the population grows, thrives, and is fully capable of maintaining itself. Adults unfed for at least three weeks were processed as IT samples ($N=5$).

All specimens used in the study were typical representatives of the corresponding species in sense of overall body structure, physiological condition (only specimens considered fully healthy were used in the study), and habitat. Culturing conditions for *L. hypogea* and gemmules of *E. fluvatilis* are well accepted and routinely used [27–29].

## RNA extraction and reverse transcription

Prior to RNA extraction, all samples were dabbed at absorbent paper for a few seconds to dispose of excessive water and lysed in 1 mL of ExtractRNA reagent (BC032, Evrogen, Moscow, Russia), homogenized with polypropylene pestle, incubated for 20 min at RT and stored at −80°C at most for two months.

RNA was extracted using the acid guanidinium thiocyanate-phenol-chloroform method [30]. The samples were then incubated with DNAse I (EN052, Thermo Fisher Scientific, Waltham, MA, USA) and the RNAse inhibitor RiboCare (EK005S, Evrogen, Moscow, Russia) for 10 min at 37°C. RNA was purified and concentrated using a 'Clean RNA Standard' spin-columns kit (BC033, Evrogen, Moscow, Russia) according to the manufacturer's protocol. The RNA concentration was measured spectrophotometrically using a NanoPhotometer (Implen, München, Germany). RNA purity was verified by A260/A280 ratio (median 2.101, range 1.75–2.667) and A260/A230 ratio (median 2.017, range 0.44–3.39). Normalization of samples within each species was performed at this stage by using equal amounts of total RNA (0.1–1 μg) for reverse transcription.

cDNA was synthesized using Random Primers from RevertAid First Strand cDNA Synthesis Kit (K1621; Thermo Fisher Scientific, Waltham, MA, USA). The required volume of RNA, 1 μL of the manufacturers' Random Hexamer Primers, and nuclease-free water (PB207S, Evrogen, Moscow, Russia) up to 12 μL were mixed. Next, 4 μL of 5x Reaction Buffer, 1 μL of RiboLock RNAse Inhibitor (20 units/μL), 2 μL of 10 mM dNTP mix, and 1 μL RevertAid M-MuLV reverse transcriptase (200 units/μL) were added per reaction. The final reaction volume was 20 μL. The tubes were gently mixed and briefly centrifuged. The reaction was incubated for 5 min at 25°C, followed by 60 min at 42°C, and terminated by heating at 72°C for 5 min. Nuclease-free water (80 μL) was added to each reaction mixture, resulting in a final volume of 100 μL. cDNA was stored at −20°C.

## Identification of target sequences and primers design

Prospective target sequences were retrieved using tblastn search [31] against the reference databases: a) *de novo* transcriptome of *L. corallorhiza*; b) genome-derived transcriptome of *H. dujardinii*; c) *de novo* transcriptome of *E. fluviatilis*; d) genome-derived transcriptome of *L. hypogea* (for details on the assemblies see Supporting Information and Table S1 in S1 File). The query amino acid sequences of *Mus musculus* were obtained using the UniProt database (accessed October-November 2024). The retrieved Porifera transcripts were translated using Expasy Translate tool [32] and

studied using web-based InterProScan v5.69-101.0 (accessed October-November 2024) to assess the conservation of the encoded protein domain structure. The identification was also validated using reciprocal pblast against the Uni-ProtKB+SwissProt databases.

For lamins, nuclear intermediate filaments, providing structural support to the nucleus and contributing to chromatin organization, we used a phylogeny-based approach. Amino acid sequences of intermediate filaments from animals other than sponges were retrieved from the UniProt database using manual or blast searches. The resulting set of amino acid sequences was aligned using MUSCLE [33] with default settings; the alignment was trimmed using TrimAl v1.4.1 [34] and used to build maximum-likelihood phylogenetic tree using IQTree web server [35]. The ML trees were built using ultrafast bootstrap (N = 1000) and SH-aLRT branch support, with automatic assessment of the optimal substitution model. The resulting trees were visualized using FigTree v1.4.4. Nucleic acid sequences of target genes are listed in the 'Nucleotide sequences from Table S2.fasta' file in S2 File(see Supporting information section).

Primer sequences were designed using the Primer–BLAST Tool (based on Primer3, version 2.5.0) with standard settings for target sequences. All primers were synthesized by Evrogen JCS (Moscow, Russia). The quality of synthesis was manufacturer-controlled by analytical electrophoresis using PAAG and mass spectral analysis. The cDNA samples used in this study were also used in x1, x10, x100, x1000, and x10000 dilutions in duplicates to create a linear regression plot for each gene (the final efficiency percentage is the mean from three independent calculations). Primer details (sequences, amplicon sizes, annealing temperatures, PCR efficiencies) are listed in Table S3 in S1 File.

### Real-time quantitative polymerase chain reaction

Real-time qPCR was performed in 96-well CFX96 Touch Real-Time PCR Detection System (Bio-Rad, Hercules, CA, USA) or in 96-well DTprime 4M1 PCR Detection System (DNA-Technology, Moscow, Russia) with SYBR Green I dye and reference dye ROX. All reagents were provided by Evrogen (Moscow, Russia) (5X qPCRmix-HS SYBR+High/LowROX, PK155L/PK156L) and the manufacturer's protocols were followed. The reaction volume was 20 µL and contained 4 µL 5x qPCRmix-HS-SYBR+High/LowROX buffer (includes 3 mM $Mg^{2+}$ ions and 0.12 mM of each dNTP per each reaction), 25 pM of each primer, and 2 µL of x5 cDNA dilution. The reaction protocol included denaturation at 95°C followed by 40 amplification cycles — 95°C for 15 s, $T_a$°C (individual for each pair of primers) for 30 s, 72°C for 30 s, — and further melting curve analysis (55–95°C, step 0.5°C). All samples were processed in triplicates. The same cDNA samples were placed repeatedly into each PCR run and served as the inter-run calibrator.

### Data analysis and statistics

Relative cDNA quantity (RQ, *i.e.,* normalized gene expression) was calculated as RQ = $CPE^{\Delta CT}$, where

- CPE = $10^{(-1/Eslope)}$ − 1; converted PCR efficiency;

- $\Delta CT$ = $Ct_n$ − $Ct_i$; averaged triplicate $Ct_n$ value for each sample, $Ct_i$ – averaged triplicate Ct value for one chosen sample.

The obtained datasets were tested for normality using the Shapiro-Wilk test. Subsequent comparisons were performed using the appropriate statistical tests. The test used is indicated in each case individually. All statistical procedures and data visualization were performed using RStudio version 4.0.3 (RStudio, USA). The significance level (α) was set at < 0.05. The following R packages were used: 'tidyverse', 'ggpubr', 'NormqPCR', 'pastecs'. The raw runs' data and R script is available in the Mendeley Data repository (see "Data availability" section).

## Results

To robustly evaluate candidate reference genes for normalization in gene expression studies of sponges, our approach was structured into two major experimental phases. In the first phase, we focused on *Leucosolenia corallorrhiza* to assess the reference gene performance under varying physiological conditions within a single species. Specifically, we

established three well-defined experimental groups reflecting the key stages of regeneration: intact tissues (IT), fragments with a growing regenerative membrane (CR; 24 hpo), and fragments with a fully sealed regenerative membrane (RM; 48 hpo). For this regenerative model cell shape transformations, and corresponding cytoskeletal rearrangements were previously described by Skorentseva et al. [36,37]. We assessed the reference gene expression stability across different biological contexts using *RHOA* as the target gene, a marker chosen because of its central role in cytoskeletal reorganization and cell motility, which are key processes in regeneration. Monitoring *RHOA* enabled us to determine whether candidate reference genes were truly suitable for normalization under various experimental challenges intrinsic to regeneration studies.

In the second phase, we extended our investigation by including three additional sponge species, *H. dujardinii*, *E. fluviatilis*, and *L. hypogea,* to test the universality of the candidate reference genes identified in *L. corallorrhiza*. By comparing these reference genes across evolutionarily and ecologically distinct species, we aimed to determine whether consistent candidate genes could serve as reliable normalizers, regardless of the taxonomic background.

### Identification of target sequences

The identification of *GAPDH*, *TBP*, *RPS3A*, *RPL13A*, *HPRT1*, and *RHOA* did not need to be supplemented by phylogenetic trees since a single copy for each gene/transcript was found in the transcriptomes. The results of reciprocal BLAST searches and the predicted domain structures of proteins encoded by the studied genes/transcripts (Fig 1a–1h and Table S2 in S1 File) unambiguously validated the identification.

For lamins, several prospective sequences were identified. All of them clearly fell into the clade containing lamins, rather than cytoplasmic intermediate filaments (Fig 2). Poriferan lamins tended to cluster together with sequences from cnidarians, and based on the topology of the tree, it seems that calcarean and demosponge lamins are highly divergent (Fig 2). We found a single lamin-coding sequence in the *L. hypogea* and *E. fluviatilis* transcriptomes. For *H. dujardinii* and *L. corallorrhiza*, we chose transcripts that encoded proteins with less predicted disorder (namely, Hdu_evm.model.contig32.21 and Lco_TRINITY_DN9368_99323) as targets for RT-qPCR.

We identified several actin-coding sequences in each database. Actin-like proteins were filtered using reciprocal blast. However, the precise phylogenetic relationships between the remaining actins remained unclear, as the attempt to construct ML trees resulted in low node support values. Thus, we aligned candidate transcripts to β- and γ -cytoplasmic isoforms of vertebrate actins; contigs Lco_TRINITY_DN24_95078 (*L. corallorrhiza*), Hdu_evm.model.28.48 (*H. dujardinii*), Lhy_evm.model.OZ017783.1.600 (*L. hypogea*), and Efl_m.16996_g.16996 (*E. fluviatilis*) encoded proteins with the highest similarity to vertebrate *ACTB* and were chosen for the subsequent analysis with *ACT1* name (Fig 3a–3c). Notably, the predicted *E. fluviatilis* sequence was fragmented and lacked the proper N-end of the amino acid chain. To validate that the identified contigs encode cytoplasmic actins in sponges, *L. corallorrhiza* regenerative membranes were immunostained with antibodies targeting various actins: monoclonal mouse anti-actin-β (BioRad MCA5775GA, clone 4C2), monoclonal mouse anti-actin-γ antibodies (BioRad MCA 5776GA, clone 2A3), and polyclonal rabbit anti-actin-β antibodies (ab8227, Abcam). Subsequent imaging revealed that cytoplasmic filamentous structures and stress fibers were stained with a certain level of differentiation between the different actin isoforms (Fig S1 in S1 File). For detailed immunostaining protocol see [36].

### Expression variability and stability assessment of candidate reference genes in *Leucosolenia corallorrhiza*

We conducted a comprehensive assessment and evaluation of candidate reference genes in *L. corallorrhiza* using four established algorithms: geNorm, NormFinder, BestKeeper, and RefFinder. Based on the analysis of existing data on the assessment of gene expression in invertebrates using real-time PCR, we selected seven candidate reference genes, *ACT1, RPL13A, GAPDH, HPRT1, RPS3A, TBP,* and *LMN1*, which are commonly used in normalization studies because of their involvement in essential cellular processes, such as cytoskeletal organization (*ACT1*), energy metabolism (*GAPDH*), ribosomal structure (*RPL13A, RPS3A*), transcription regulation (*TBP*), nucleotide metabolism (*HPRT1*), and nuclear integrity (*LMN1*).

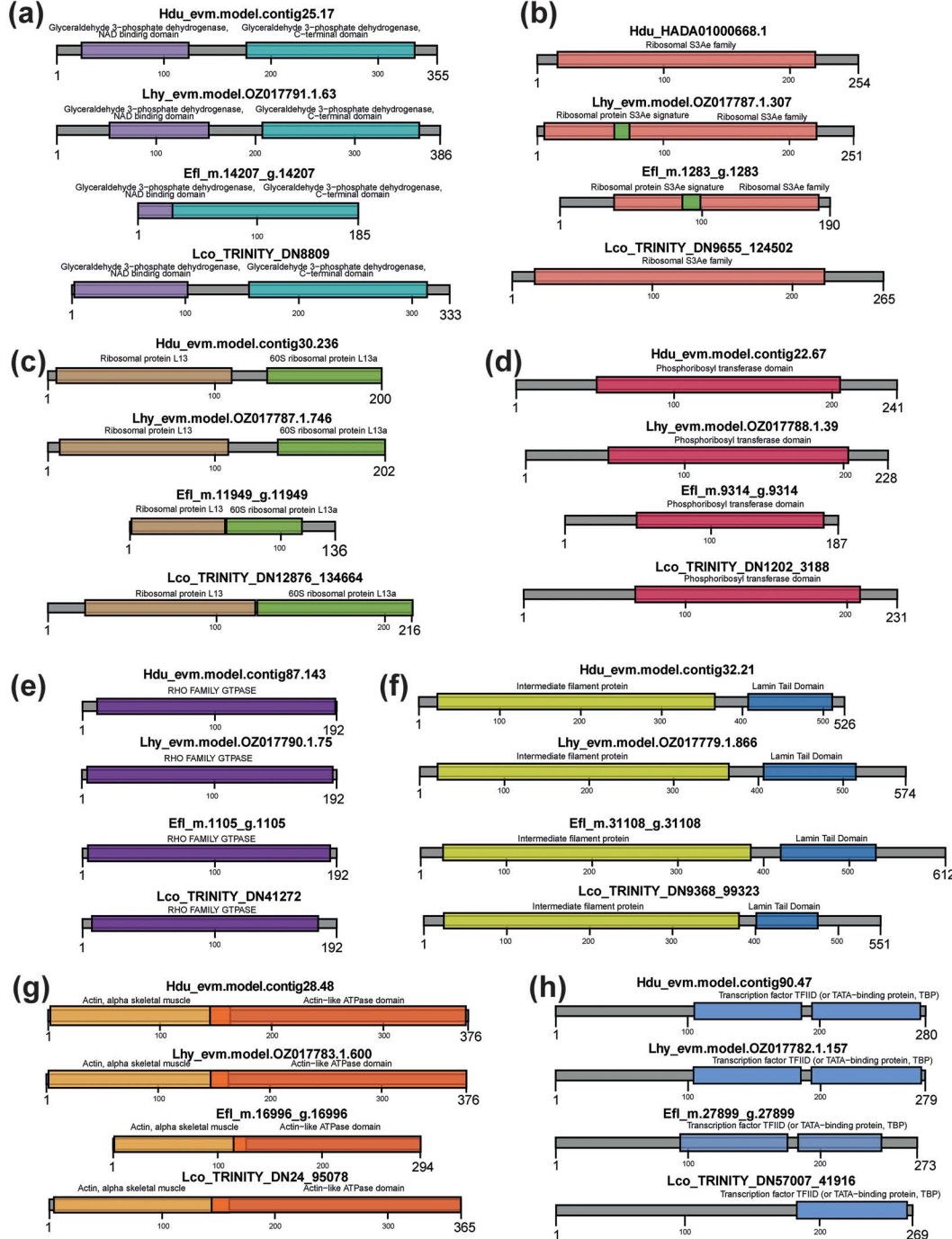

**Fig 1. Domain structures of proteins encoded by the identified transcripts.** GAPDH **(a)**, RPS3A **(b)**, RPL13A **(c)**, HPRT1 **(d)**, RHOA **(e)**, LMN1 **(f)**, ACT1 **(g)**, TBP **(h)**. Hdu – *Halisarca dujardinii*, Lhy – *Lycopodina hypogea*, Efl – *Ephydatia fluviatilis*, Lco – *Leucosolenia corallorrhiza*.

To assess the variability in the expression of the selected genes, we first analyzed their cycle threshold (Ct) values ($N$ = 14). Expression analysis of the seven candidate reference genes across all samples from regeneration experiment on *L. corallorrhiza* revealed significant differences in both expression levels and variability (Fig 4a and Table 1). The

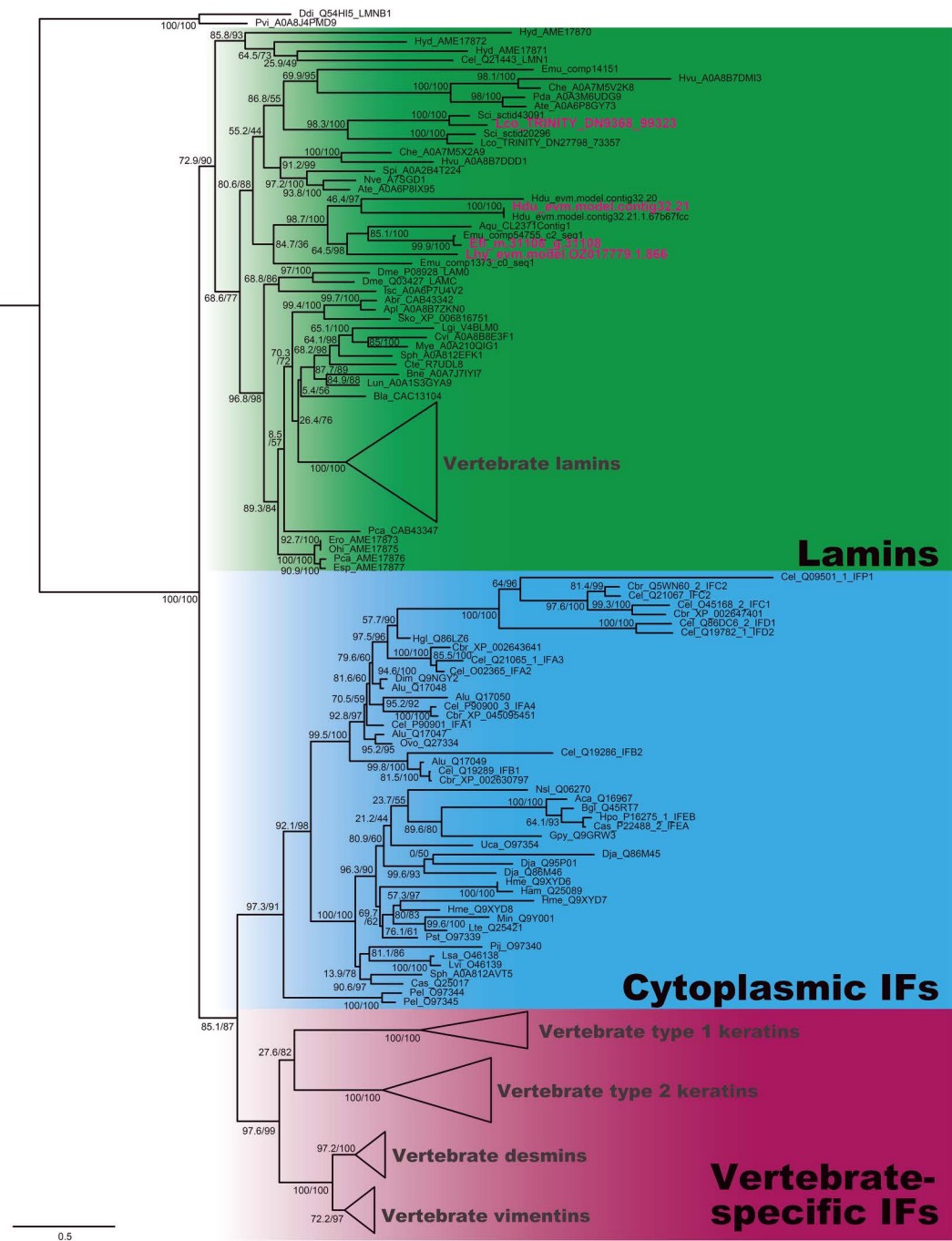

**Fig 2. Phylogenetic relationships among intermediate filament (IF) proteins.** A maximum likelihood tree was constructed using IQTree with ultrafast bootstrap (N = 1000) and SH–aLRT branch support. A full-length alignment trimmed with TrimAl was used to construct the tree. The substitution model LG + G4 was chosen automatically. Node labels represent ultrafast bootstrap/SH–aLRT support values. Names of sponge sequences used for PCR analysis in the study are highlighted in magenta. Abr – *Astropecten brasiliensis*, Aca – *Aplysia californica*, Apl – *Acanthaster plancii*, Aqu – *Amphimedon queenslandica*, Ate – *Actinia tenebrosa*, Alu – *Ascaris lumbricoides*, Aam – *Amphibalanus amphitrite*, Bgl – *Biomphalaria glabrata*, Bne – *Bugula neritina*, Cas – *Cornu aspersum*, Cel – *Caenorhabditis elegans*, Cbr – *Caenorhabditis briggsae*, Che – *Clytia hemisphaerica*, Cin – *Ciona intestinalis*, Csa – *Ciona savignyi*, Cte – *Capitella teleta*, Cvi – *Crassostrea virginica*, Ddi – *Dictyostelium discoideum*, Dme – *Drosophila melanogaster*, Dpu – *Daphnia pulex*, Dim – *Dirofilaria immitis*, Dja – *Dugesia japonica*, Emu – *Ephydatia muelleri*, Efl – *Ephydatia fluviatilis*, Ero – *Euperipatoides rowelli*, Esp – *Eoperipatus* sp. LH–2012, Gpy – *Glottidia pyramidata*, Ham – *Haemopis marmorata*, Hex – *Hypsibius dujardini*, Hdu – *Halisarca dujardinii*, Hgl – *Heterodera*

glycines, Him – *Hirudo medicinalis*, Hpo – *Helix pomatia*, Hvu – *Hydra vulgaris*, Isc – *Ixodes scrapularis*, Lco – *Leucosolenia corallorrhiza*, Lgi – *Lottia gigantea*, Lun – *Lingula unguis*, Lte – *Lumbricus terrestris*, Lhy – *Lycopodina hypogea*, Lsa – *Lineus sanguineus*, Lvi – *Lineus viridis*, Min – *Myxicola infundibulum*, Mye – *Mizuhopecten yessoensis*, Nve – *Nematostella vectensis*, Nsl – *Nototodarus sloanii*, Ohi – *Ooperipatus hispidus*, Osp – *Oscarella* sp., Ovo – *Onchocerca volvulus*, Pel – *Parasagitta elegans*, Phi – *Principapillatus hitoyensis*, Pca – *Priapulus caudatus*, Pij – *Phoronis ijimai*, Pst – *Phascolion strombus*, Pvi – *Polysphondylium violaceum*, Sci – *Sycon ciliatum*, Spi – *Stylopora pistillata*, Sph – *Sepia pharaonis*, Sko – *Saccoglossus kowalewski*, Uca – *Urechis caupo*.

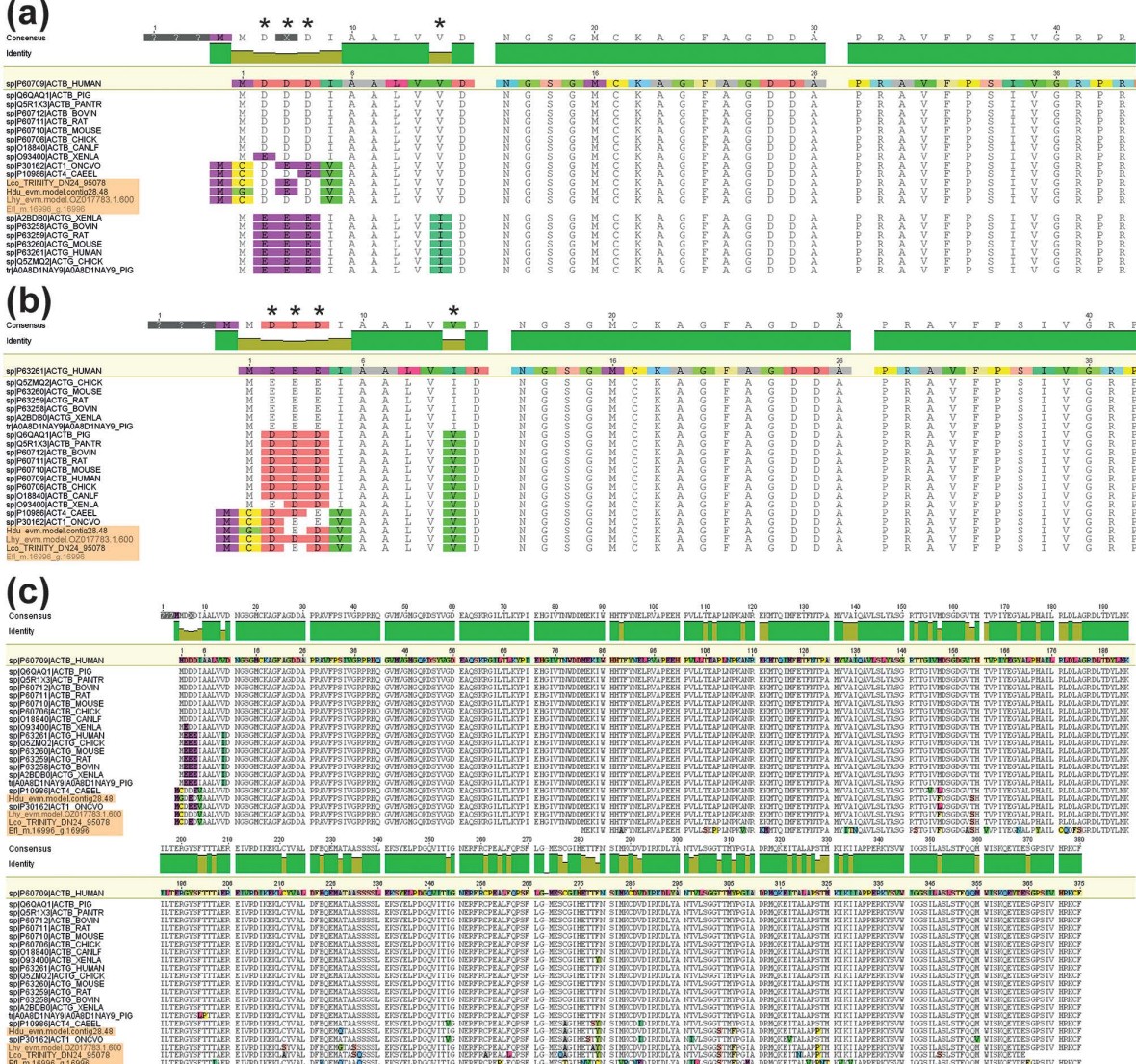

**Fig 3. Alignment of amino acid sequences encoded by the identified transcripts. (a, b)** – N-end of the alignment with human β-cytoplasmic actin and γ-cytoplasmic actin set as reference sequences; **(c)** – the whole alignment with human β-cytoplasmic actin set as the reference. Colored residues outside the reference sequence represent residues that deviate from the reference sequence. Asterisks indicate amino acid positions essential for distinguishing between β- and γ-actin in vertebrates.

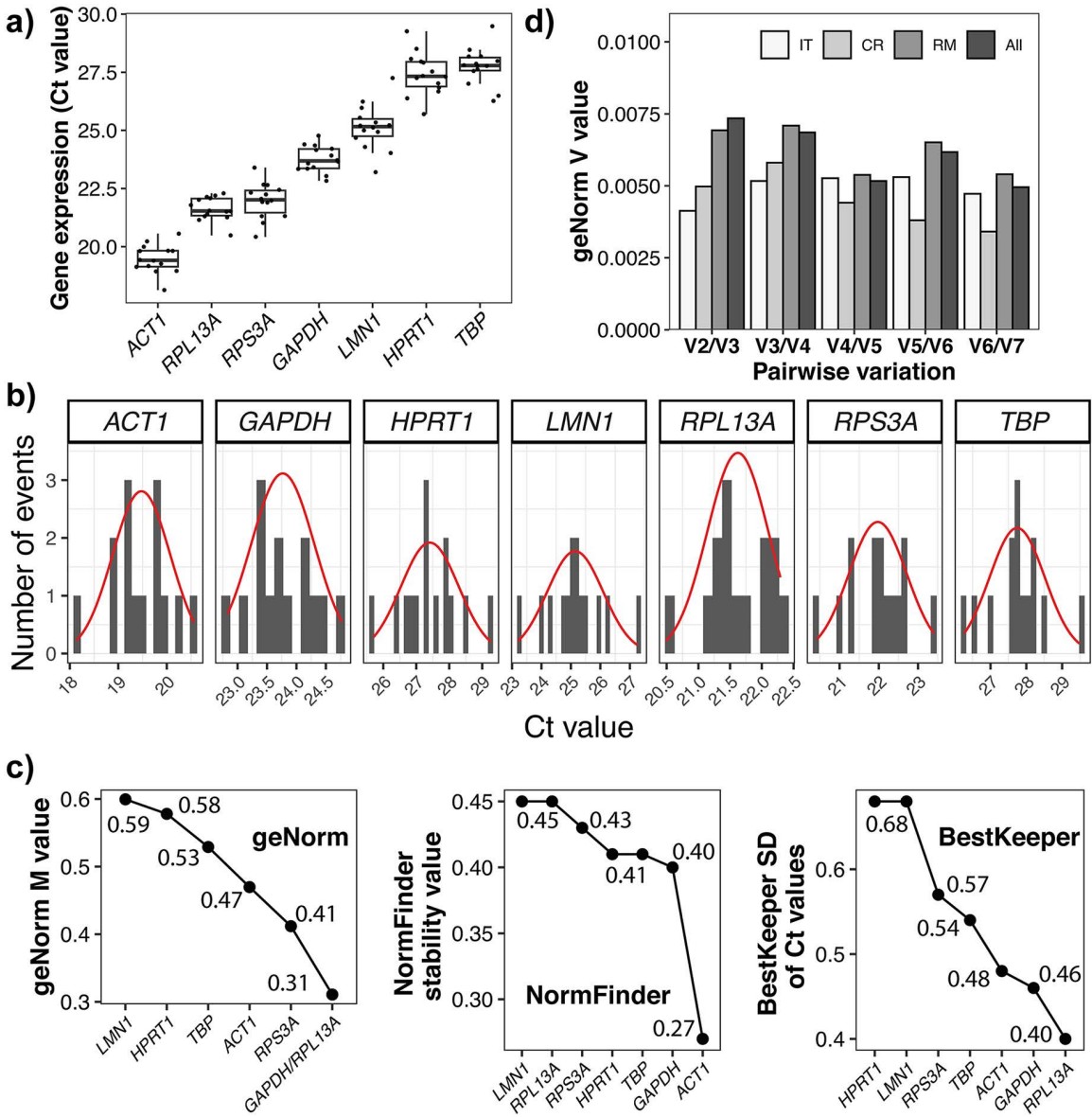

**Fig 4. Expression variability and stability assessment of candidate reference genes in *Leucosolenia corallorrhiza*.** (a) Boxplots showing the distribution of Ct values for the seven candidate reference genes across all samples, with lower Ct values indicating higher transcript abundance. Upper whisker extends from the top of the box (third quartile, Q3) to the most extreme data point within 1.5 times the Interquartile Range (IQR) above Q3. Lower whisker extends from the bottom of the box (first quartile, Q1) to the most extreme data point within 1.5 times the IQR below Q1. (b) Histograms with fitted curves (red) illustrating the frequency distribution of the Ct values for each gene. (c) Stability ranking of the reference genes calculated using four algorithms: geNorm (M values), NormFinder (stability values), and BestKeeper (standard deviation of $C_t$-values). Lower values correspond to higher stabilities. (d) Pairwise variation (V values) between combinations of reference genes across experimental groups (IT, intact tissue; CR, 'crescent'/growing regenerative membrane; RM, fully sealed regenerative membrane; All, total dataset), as calculated using geNorm.

coefficients of variation (CV) for all genes remained below 0.04, suggesting relatively stable expression in biological replicates. The lowest Ct values, corresponding to the highest expression, were observed for *ACT1*, whereas *TBP* exhibited the highest Ct values, indicating a lower transcript abundance. *RPL13A* and *RPS3A* showed intermediate and relatively stable expression, as reflected by their narrow interquartile ranges and limited number of outlier points. In contrast,

**Table 1. Cycle threshold (Ct) values and descriptive statistics for 7 candidate endogenous referent genes assayed across 14 samples from *Leucosolenia corallorrhiza* tissues.**

| Gene (*Lco*-) | Mean Ct | SD | CV | Min Ct | Median Ct | Max Ct | Range Ct | *Shapiro-Wilk. stat.* | *p-value* |
|---|---|---|---|---|---|---|---|---|---|
| ACT1 | 19.48 | 0.62 | 0.031 | 18.14 | 19.42 | 20.56 | 2.42 | 0.974 | 0.923 |
| RPL13A | 21.62 | 0.50 | 0.023 | 20.49 | 21.54 | 22.29 | 1.81 | 0.937 | 0.385 |
| GAPDH | 23.77 | 0.57 | 0.023 | 22.83 | 23.68 | 24.77 | 1.93 | 0.978 | 0.959 |
| HPRT1 | 27.41 | 0.91 | 0.033 | 25.7 | 27.32 | 29.26 | 3.56 | 0.991 | 0.999 |
| RPS3A | 21.98 | 0.76 | 0.035 | 20.42 | 22.02 | 23.40 | 2.98 | 0.975 | 0.931 |
| TBP | 27.74 | 0.80 | 0.029 | 26.27 | 27.79 | 29.48 | 3.21 | 0.938 | 0.389 |
| LMN1 | 25.14 | 0.98 | 0.039 | 23.20 | 25.16 | 27.25 | 4.05 | 0.974 | 0.921 |

*HPRT1* and *TBP* demonstrated pronounced variability, with more outliers (in case of *TBP*), suggesting less consistent expression across the samples. Next, we plotted histograms with fitted curves to show the distribution of the Ct values for each gene (Fig 4b). *ACT1*, *RPL13A*, and *GAPDH* displayed approximately normal (unimodal and symmetric) Ct value distributions, indicating uniform gene expression within the dataset. Meanwhile, *HPRT1* and TBP showed more skewed or multimodal distributions, reflecting greater instability and potential condition-dependent expression fluctuations.

To evaluate the stability of the selected reference genes, we performed an analysis using the geNorm, NormFinder and BestKeeper applications (Fig 4c). The geNorm algorithm evaluates the stability of the reference gene by calculating the M value, which represents the average pairwise variation of a gene relative to all the other tested genes [5]. Genes with lower M values are considered more stable. Our results showed that *RPL13A, RPS3A* and *GAPDH* were among the most stably expressed, whereas *HPRT1* and *LMN1* exhibited the highest variability. In addition to ranking genes, geNorm calculates pairwise variation (V value, the average variation between the normalization factors using n and n + 1 reference genes, indicating the effect of adding an extra gene on normalization accuracy) to determine the minimum number of genes required for reliable normalization. The algorithm assumes that the most stably expressed genes should have minimal variation relative to each other, with a recommended threshold of 0.15 [5]. The analysis of pairwise variation coefficients demonstrated consistently low V values across all experimental groups (Fig 4d and Table S4 in S1 File). The V2/3 values for all conditions, including intact tissues, growing regenerative membranes, and fully sealed regenerative membranes, were well below the 0.15 threshold, indicating that the use of only two reference genes was sufficient for reliable normalization. This confirms that the selected reference genes exhibit strong co-expression stability and that further inclusion of additional genes does not significantly improve the normalization accuracy.

The NormFinder algorithm ranks candidate reference genes based on their stability value, which is determined by estimating intragroup (within experimental conditions) and intergroup (between conditions) variations [22]. Unlike geNorm, which relies on pairwise comparisons, NormFinder calculates the systematic variance of each gene independently and considers the effects of experimental grouping. Genes with lower stability values were more reliable for normalization. According to NormFinder, *ACT1* was identified as the most stable gene, followed by *GAPDH* and *TBP*, whereas *RPL13A* and *LMN1* showed higher variability (Fig 4c and Table S5 in S1 File).

The BestKeeper algorithm assesses reference gene stability based on their standard deviation (SD) and coefficient of variance (CV). Genes with lower SD and CV were considered more stable [23]. Unlike geNorm and NormFinder, BestKeeper does not assume equal expression ratios between genes but instead evaluates absolute Cq values directly. In our analysis, *GAPDH*, *RPL13A,* and *ACT1* had the lowest SD values, suggesting that they were the most stable genes according to BestKeeper, whereas *HPRT1* and *LMN1* had the highest SD values, indicating significant variability (Fig 4c). Given that *RPL13A*, *ACT1,* and *GAPDH* consistently appeared among the top three across three independent algorithms (ranked top-3 at least by two of the three algorithms) (see Table 2), and considering their normal distribution and low

**Table 2. Stability ranking of candidate reference genes in sponge species according to GeNorm, NormFinder, and BestKeeper analysis.**

| Species | Ranking | 1 | 2 | 3 | Ranked top-3 at least in 2 of 3 algorithms |
|---|---|---|---|---|---|
| *Leucosolenia corallorrhiza* | GeNorm | *GAPDH/RPL13A* | *RPS3A* | *ACT1* | *RPL13A, ACT1, GAPDH* |
| | NormFinder | *ACT1* | *GAPDH* | *TBP* | |
| | BestKeeper | *RPL13A* | *GAPDH* | *ACT1* | |
| *Halisarca dujardinii* | GeNorm | *RPS3A/HPRT1* | *ACT1* | *RPL13A* | *RPS3A, RPL13A, HPRT1* |
| | NormFinder | *RPL13A* | *RPS3A* | *HPRT1* | |
| | BestKeeper | *TBP* | *RPL13A* | *RPS3A* | |
| *Ephydatia fluviatilis* | GeNorm | *GAPDH/ACT1* | *LMN1* | *RPS3A* | *GAPDH, ACT1, RPS3A* |
| | NormFinder | *GAPDH* | *ACT1* | *RPS3A* | |
| | BestKeeper | *RPS3A* | *ACT1* | *HPRT1* | |
| *Lycopodina hypogea* | GeNorm | *GAPDH/RPL13A* | *ACT1* | *RPS3A* | *GAPDH, RPL13A, RPS3A, ACT1* |
| | NormFinder | *RPS3A* | *HPRT1* | *GAPDH* | |
| | BestKeeper | *RPL13A* | *ACT1* | *LMN1* | |

variability in expression, using these genes provides robust and reliable normalization across different experimental conditions in *Leucosolenia corallorrhiza*.

## Cross-species evaluation of reference gene stability

To assess the universality and robustness of the identified candidate reference genes, we extended our analysis to three additional sponge species representing diverse taxonomic and ecological backgrounds: *Halisarca dujardinii* (marine demosponge), *Ephydatia fluviatilis* (freshwater demosponge), and *Lycopodina hypogea* (carnivorous marine demosponge). This comparative approach enabled us to determine whether consistent reference genes could serve as reliable normalizers across different sponge taxa, thereby providing insights into the broader applicability of our findings for quantitative gene expression studies in Porifera.

In *H. dujardinii*, geNorm identified *RPS3A* and *HPRT1* as the most stable genes, *ACT1* and *RPL13A* also demonstrated high stability. NormFinder ranked *RPL13A*, *RPS3A*, and *HPRT1* as the top candidates, whereas BestKeeper analysis confirmed the high stability of *RPL13A, RPS3A*, and *TBP*. Collectively, *RPS3A, RPL13A,* and *HPRT1* were the most reliable reference genes for this species (Fig 5a and Table 2).

Both geNorm and NormFinder indicated that *GAPDH, ACT1,* and *RPS3A* were the most stably expressed genes in *E. fluviatilis*. The BestKeeper analysis was consistent, highlighting *RPS3A, ACT1,* and *HPRT1* as the top performers. Thus, *ACT1, RPS3A,* and *GAPDH* were determined as the most suitable reference genes for this species (Fig 5b and Table 2).

For *L. hypogea*, geNorm suggested *GAPDH* and *RPL13A* as the most stable pair, followed by *ACT1* and *RPS3A*. NormFinder ranked *RPS3A*, *HPRT1*, and *GAPDH* as the most stable genes, while BestKeeper analysis favored *RPL13A*, *ACT1*, and *LMN1*. Based on these results, *GAPDH*, *RPL13A*, *RPS3A*, and *ACT1* were identified as the most stable genes in *L. hypogea* (Fig 5c and Table 2).

To compare our multi-algorithm approach with a commonly used consensus tool, we also analyzed candidate reference genes using RefFinder. The resulting rankings are shown in Fig S2 in S1 File. RefFinder integrates the outputs of four commonly used algorithms (geNorm, NormFinder, BestKeeper, and the ΔCt method) by calculating their geometric mean to generate a consensus stability ranking of candidate reference genes [24]. The RefFinder consensus rankings of candidate reference genes showed some species-specific variation and, in several cases, diverged from the stability patterns obtained with individual algorithms. In *L. corallorrhiza*, RefFinder placed *RPL13A* and *ACT1* at the top, while *RPS3A* appeared least stable. However, in our multi-algorithm evaluation, both *RPS3A* and *RPL13A* were consistently among the most stable genes.

 

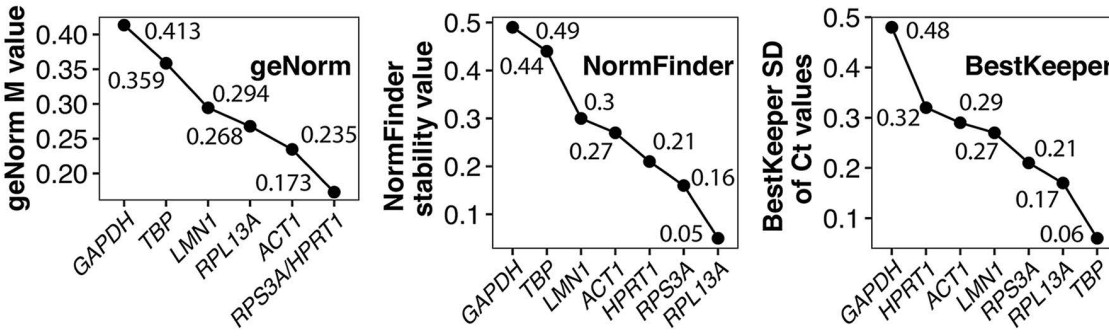

**a)** *Halisarca dujardinii*

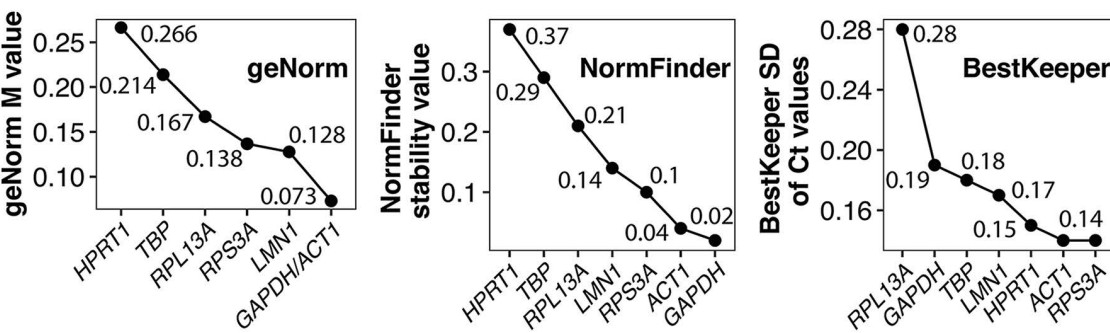

**b)** *Ephydatia fluviatilis*

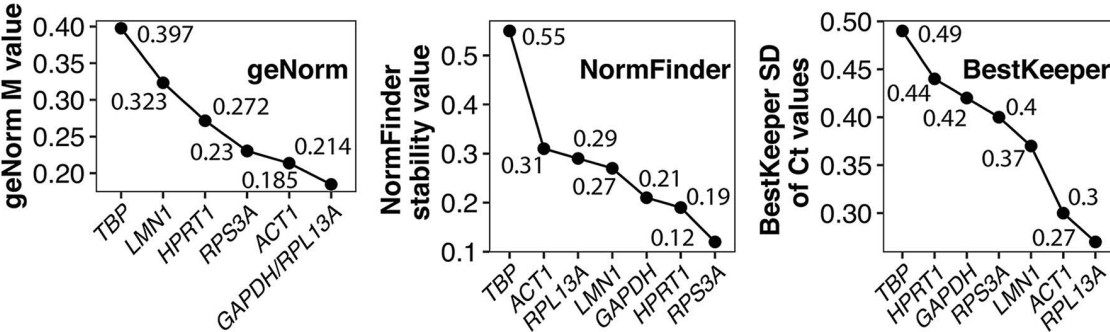

**c)** *Lycopodina hypogea*

**Fig 5. Stability ranking of candidate reference genes in the three sponge species based on three statistical algorithms.** The expression stability of the seven candidate reference genes was evaluated in **(A)** *Halisarca dujardinii*, **(B)** *Ephydatia fluviatilis*, and **(C)** *Lycopodina hypogea* using geNorm (first column), NormFinder (second column), and BestKeeper (third column). Each graph shows the stability values or ranking scores for the tested genes, with lower values indicating a higher expression stability.

In *H. dujardinii*, RefFinder ranked *LMN1* and *RPL13A* as the most stable genes. This contrasted all other outputs, as none of the algorithms ranked *LMN1* among the most stable.

In *E. fluviatilis*, RefFinder identified *TBP* as the most stable genes, but surprisingly placed *RPL13A* and *HPRT1* at the very bottom. This was biologically implausible, since *TBP* displayed relatively high variability across all other algorithms,

and none of the independent methods supported such instability. These comparisons demonstrate that RefFinder may over- or under-estimate the stability of certain genes, producing rankings that contradict other algorithmic outputs.

Taken together, comparative analysis across all three species indicated that the most consistently stable reference genes included a ribosomal gene (*RPL13A* or *RPS3A*), *ACT1*, and *GAPDH*. This combination provides a reliable basis for normalization in gene expression studies involving diverse sponge species (Table 2).

**Effect of single vs. multiple reference gene normalization on *RHOA* expression**

Using the three selected reference genes individually for normalization resulted in notably different *RHOA* expression values, whereas the combined normalization factor (NF) yielded more consistent measurements. As shown in Fig 6a, single-gene normalization can over- or underestimate *RHOA* levels for the same sample. For example, the *RHOA* expression normalized to *ACT1* was higher than that normalized to *GAPDH* or *RPL13A*. Such disparities indicate how much using a single reference gene can bias the results. In contrast, when using the combined NF, the *RHOA* expression of each sample fell between the extremes of the single-gene values, resulting in a more moderate and uniform estimate (see Table S6 in S1 File). This pattern was consistent across all experimental conditions in the regeneration assay for *L. corallorrhiza*.

The scatter plot panels reinforce that normalization with multiple reference genes reduces variability. For *L. corallorrhiza* (Fig 6b), normalization to a single gene, even to a commonly used and relatively stable gene such as *ACT1*, results in greater variability and can lead to significant differences in gene expression between treatments. In *L. hypogea*, the expression values for the same treatment can differ by up to 1.5-fold, depending on the reference gene used (Fig 6c). This variability reflects the sensitivity of single-gene normalization to inter-sample variation and occasional outliers. In contrast, normalization with combined NF produces values that exhibit significantly smaller standard deviations (see Table 3). Similar trends were evident in *H. dujardinii* and *E. fluviatilis*: the single-reference normalized values varied considerably, whereas the NF-normalized values clustered tightly, indicating consistent expression.

## Discussion

Gene expression analysis has long become an essential tool in contemporary biological research, providing insights into molecular mechanisms across a wide range of organisms. Although significant progress has been made in developing methods for assessing gene expression, most protocols and normalization strategies have been optimized for model organisms or specific experimental setups. Extending these approaches to complex and evolutionarily distinct groups, such as Porifera (sponges), presents unique challenges. In particular, the selection and validation of appropriate normalization methods and reference genes remain problematic, as much of the literature is dedicated to single-object studies or relies on housekeeping genes, whose universality is questionable in such distant taxa [38–41]. Therefore, addressing gene expression in Porifera requires a critical re-examination of methodological strategies, ensuring that analyses adequately reflect the evolutionary and morphological diversity of this group, rather than being constrained by the limitations of standard single-gene approaches.

In non-model phylogenetic groups, difficulties with gene expression normalization often begin at the first stage identifying correct candidate reference gene sequences. Errors in this step may result in the selection of incomplete, misannotated, or paralogous sequences that are not true functional orthologs of the intended genes. This, in turn, can distort RT-qPCR results, as the apparent expression stability of such sequences may reflect artifacts of incorrect identification rather than genuine biological consistency. In several studies on invertebrates and other non-model taxa [42–45], detailed sequence validation was not explicitly discussed, and BLAST search results were used directly without additional confirmation of orthology, domain structures, or phylogenetic placement. Such a simplified approach can lead to the misinterpretation of normalized data, particularly in groups where gene families are highly divergent, contain multiple copies, and have taxon-specific isoforms. Therefore, comprehensive verification, including phylogenetic analysis, domain structure assessment, and conserved motif searches, is essential for accurate selection of reference genes in non-model

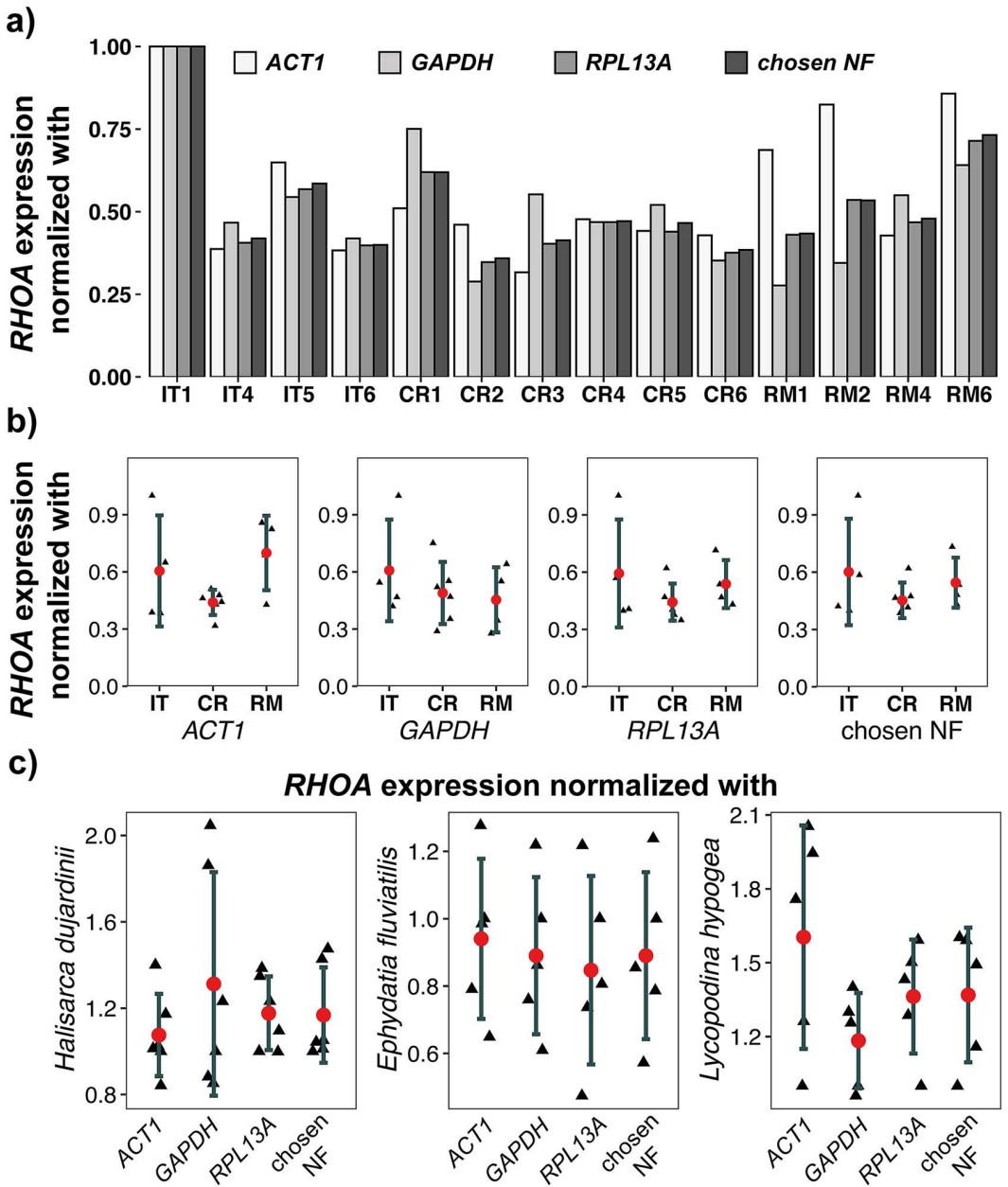

**Fig 6. Impact of normalization strategy on *RHOA* expression estimated across sponge species. (a)** Normalized *RHOA* gene expression in 3 experimental groups of regeneration model (*L. corallorrhiza*); **(b)** scatter plots showing normalized *RHOA* mRNA expression for individual experiments in regeneration model of *L. corallorrhiza*; red dots represent mean value; whiskers represent standard deviation (SD); **(c)** Normalized *RHOA* mRNA expression for 3 other sponge species. Chosen NF is normalization factor, calculated as geometric mean of relative quantities of *GAPDH*, *ACT1* and *RPL13A*. Normalization was performed according to Material & Methods section, IT1 sample was chosen as a control one.

organisms. Another important aspect of our study is the reliable identification of target sequences due to the combination of reciprocal blast search and domain structure prediction. This approach allowed us to: 1) validate if the target sequences are indeed evolutionary conserved in early-branching metazoans such as sponges; 2) distinguish between closely related paralogous genes/transcripts differing in the domain architectures; 3) filter out possible contaminants. This approach was

**Table 3. Normalized *RHOA* mRNA expression for individual experiments in regeneration model of *L. corallorrhiza* (as shown in Fig 6b). Mean±SD. Note the instability of results in case of normalization using only one gene.**

| | | Normalized *RHOA* expression in *Lco* | | | |
|---|---|---|---|---|---|
| | Normalized with: | *ACT1* | *GAPDH* | *RPL13A* | combined chosen NF |
| Condition: | IT | 0.605±0.291 | 0.608±0.266 | 0.593±0.282 | 0.601±0.279 |
| | CR | 0.439±0.066 | 0.489±0.163 | 0.443±0.097 | 0.45±0.093 |
| | RM | 0.699±0.195 | 0.453±0.17 | 0.537±0.126 | 0.545±0.131 |
| | | Normalized *RHOA* expression in other species | | | |
| | Normalized with: | *ACT1* | *GAPDH* | *RPL13A* | combined chosen NF |
| Species: | *Hdu* | 1.076±0.191 | 1.312±0.518 | 1.177±0.029 | 1.168±0.221 |
| | *Efl* | 0.94±0.238 | 0.89±0.233 | 0.847±0.28 | 0.89±0.248 |
| | *Lhy* | 1.604±0.454 | 1.183±0.194 | 1.363±0.232 | 1.369±0.273 |

particularly important given the high level of sequence divergence and the presence of gene duplications in early-branching metazoans. Furthermore, identification of conserved regions enabled the design of species-specific primers with improved specificity and efficiency.

Thus, the combination of evolutionary-informed gene selection and experimental validation provides a more robust framework for reference gene identification in non-model organisms. This strategy not only increases the reliability of RT-qPCR normalization but also enhances the reproducibility and comparability of gene expression studies across species. We propose that such an approach should be considered a recommended practice, particularly for taxa with limited genomic annotation and high evolutionary diversity, such as Porifera.

Initial screening of *L. corallorrhiza* identified ribosomal protein genes (*RPL13A* and *RPS3A*) as well as *GAPDH* and *ACT1* as the most stable candidates. Ribosomal genes are often recommended for normalization because of their fundamental roles in protein synthesis and their presumed consistent expression. Similarly, traditional housekeeping genes such as *ACT1* and *GAPDH* demonstrated stable performance in our dataset. However, previous research in other metazoans has shown that the expression of these genes can be highly context-dependent and may vary with tissue type, developmental stage, or experimental treatment [46–48]. Our findings confirm that, while such genes are suitable for specific experimental setups, such as regeneration in *L. corallorrhiza,* they should not be assumed to be universally stable.

Three commonly used algorithms, geNorm, NormFinder and BestKeeper, were employed to assess expression stability. Similar to other studies, some discrepancies in gene rankings were observed across the algorithms. For example, while NormFinder ranked *ACT1* as the most stable gene, geNorm ranked it lower. These inconsistencies underscore the importance of integrated approaches. We also tested RefFinder as a consensus-based ranking tool but ultimately decided not to include its outputs in the final results. During our analysis, we observed that RefFinder sometimes produced biologically implausible rankings. For example, in *Ephydatia fluviatilis*, it ranked *TBP* as the most stable gene, despite this gene showing relatively high variability across all other stability assessments (Figs 5b and S2 in S1 File). Similar issues have been documented previously, for instance, de Spiegelaere et al. [49] reported that RefFinder does not take PCR amplification efficiencies into account, which can bias the rankings. This limitation is particularly critical in non-model species, where PCR efficiency can vary significantly between targets due to sequence divergence, and uncorrected data may therefore lead to misleading conclusions.

Our analysis further demonstrated that normalization using a combination of three genes, namely *RPL13A*, *ACT1*, and *GAPDH*, provides robust and reproducible normalization in *L. corallorrhiza*. These results are consistent with the MIQE (Minimum Information for Publication of Quantitative Real-Time PCR Experiments) guidelines, which recommend the use of multiple validated reference genes [1,5].

Extension of the analysis to three additional sponge species revealed considerable interspecies variability in reference gene stability. No single gene was universally stable across all of the four species. For example, *RPL13A*, highly ranked in *L. corallorrhiza*, showed lower stability in *H. dujardinii*, whereas *TBP*, unstable in *L. corallorrhiza*, ranked higher in other species. Such interspecies variations in gene expression stability should be taken in account during experiment planning.

Accordingly, reference genes should be selected and validated individually for each model species rather than extrapolated across related taxa. Based on our results, it appears unlikely that a universal reference gene panel applicable to all sponge species can be established. Instead, this study provides a general methodological framework for the systematic identification and validation of candidate reference genes, rather than proposing definitive universal recommendations. We consider that the reference gene stability in sponges is primarily species-specific rather than determined by ecological or functional classification. This is consistent with previous studies in other non-model organisms, where reference gene performance has been shown to depend on species-specific transcriptional regulation, physiological state, and experimental conditions [50–53].

The consequences of the reference gene selection were further illustrated by our analysis of *RHOA* expression. *RHOA* is a key regulator of cytoskeletal remodeling and cell motility [54], making it a biologically relevant target during regeneration. However, when normalized to a single reference gene, even to a relatively stable one such as *ACT1,* we observed significant variability across samples and treatments, which could result in false-positive or false-negative conclusions. In contrast, normalization using the geometric mean of *RPL13A, ACT1*, and *GAPDH* mRNA quantities significantly reduced variability and yielded more consistent and biologically meaningful expression patterns.

It should also be noted that the present study was conducted under baseline conditions and does not account for potential changes in gene expression stability under environmental or physiological stress. Numerous studies have demonstrated that commonly used reference genes may vary in their expression under conditions such as thermal stress, nutrient deprivation, or exposure to toxic compound [55–58].

Therefore, while our results provide a set of robust candidate genes (e.g., *RPL13A*, *ACT1*, *GAPDH*, and *RPS3A*) that perform consistently across species under standard conditions, their use in stress-related experiments should be preceded by condition-specific validation. In this context, the gene panel identified in this study can serve as a rational starting point for selecting and testing reference genes in future investigations of sponge stress physiology and environmental responses.

## Limitations

Several factors should be considered when interpreting the results of this study. First, although RNA integrity and purity were carefully monitored, the inherent challenges of RNA extraction from sponge tissues — such as the presence of mineral skeleton components and polysaccharides — may have introduced variability in template quality. Second, for some experimental groups, the number of biological replicates was limited by specimen availability, as well as geographic limitations in sampling, potentially reducing the robustness of statistical comparisons. Third, while the use of multiple validated reference genes improves normalization accuracy, it also increases the experimental workload and resource demands, which may limit the feasibility of this approach in large-scale or time-sensitive studies.

Finally, the stability of reference genes can be influenced by the specific experimental condition and tissue type analyzed; therefore, the suitability of the selected genes should be re-evaluated when experimental designs deviate substantially from those tested here.

## Conclusion

We identified robust reference gene combinations for RT-qPCR normalization in diverse sponge species, emphasizing the need for sequence-level verification and multi-algorithm stability assessment. In *Leucosolenia corallorrhiza*, *RPL13A, ACT1,* and *GAPDH* provided the most consistent performance, and similar patterns were observed in other species.

Multi-gene normalization markedly reduced variability and prevented misleading interpretations, as demonstrated for *RHOA* expression. These results offer a validated framework for accurate and reproducible gene expression studies in Porifera, particularly in ecologically significant ones aimed at understanding ecosystem-level responses to environmental stressors, including thermal stresses, pollution, and other anthropogenic impacts.

## Supporting information

**S1 File. Contains methods' clarifications and additional tables; figure and table legends are included within the file.**
(DOCX)

**S2 File. Nucleotide sequences identified and used for PCR analysis in the study.**
(FASTA)

## Acknowledgments

The authors acknowledge the support of the Lomonosov Moscow State University Program of Development (DTprime 4M1 PCR). KS acknowledges Ostrogradski scholarship 2024 (French Embassy in Russia) in frame of which studies of L. hypogea were conducted in the University of Montpellier. The authors sincerely thank Julia Kraus ant Anton Bogomolov (Koltzov IDB RAS), Emilie Le Goff, Nelly Godefroy and Stephen Baghdiguian (Université Montpellier, France) for continuous assistance regarding sponge maintenance and helpful advice. The authors are grateful to the Core Facility Centrum of the Koltzov Institute of Developmental Biology RAS supported by the IDB RAS research program №0088-2024-0012 for the possibility to use the equipment. Conflict of interests The authors declare that they have no conflict of interest. Funding The research was supported by the Russian Science Foundation project no. 23–74–10005 (studies in Leucosolenia corallorrhiza and Halisarca dujardinii) and no. 24-14-00452 (studies in Ephydatia fluviatilis and Lycopodina hypogea). The funders had no role in study design, data collection and analysis, decision to publish, or preparation of the manuscript.

## Author contributions

**Conceptualization:** Kseniia V. Skorentseva, Andrey I. Lavrov, Aleena A. Saidova.

**Formal analysis:** Kseniia V. Skorentseva, Nikolai P. Melnikov, Andrey I. Lavrov, Aleena A. Saidova.

**Funding acquisition:** Andrey I. Lavrov.

**Investigation:** Kseniia V. Skorentseva, Nikolai P. Melnikov.

**Methodology:** Kseniia V. Skorentseva, Nikolai P. Melnikov, Aleena A. Saidova.

**Supervision:** Andrey I. Lavrov, Aleena A. Saidova.

**Validation:** Aleena A. Saidova.

**Visualization:** Kseniia V. Skorentseva, Nikolai P. Melnikov.

**Writing – original draft:** Kseniia V. Skorentseva, Aleena A. Saidova.

**Writing – review & editing:** Alexander V. Ereskovsky, Andrey I. Lavrov, Aleena A. Saidova.

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
