## [Decision Letter · Decision Letter 0]

22 Feb 2026

PONE-D-26-03346Validating the classics: accurate reference gene panel for reliable RT-qPCR in PoriferaPLOS One

Dear Dr. Saidova,

Thank you for submitting your manuscript to PLOS ONE. After careful consideration, we feel that it has merit but does not fully meet PLOS ONE’s publication criteria as it currently stands. Therefore, we invite you to submit a revised version of the manuscript that addresses the points raised during the review process.

We look forward to receiving your revised manuscript.

Kind regards,

Fang Zhu, Ph.D.

Academic Editor

PLOS One

Journal Requirements:

“The research was supported by the Russian Science Foundation project no. 23–74–10005 (studies in Leucosolenia corallorrhiza and Halisarca dujardinii) and no. 24-14-00452 (studies in Ephydatia fluviatilis and Lycopodina hypogea).”

3. In the online submission form, you indicated that “A draft dataset supporting the conclusions of this study is available upon request. The R script used to analyze and visualize the data is also available.”

5. Please upload a new copy of Figure 2 as the detail is not clear. Please follow the link for more information:  https://journals.plos.org/plosone/s/figures

6. We notice that your supplementary [figures/tables] are included in the manuscript file. Please remove them and upload them with the file type 'Supporting Information'. Please ensure that each Supporting Information file has a legend listed in the manuscript after the references list.

Reviewers' comments:

Reviewer's Responses to Questions

**Comments to the Author**

1. Is the manuscript technically sound, and do the data support the conclusions?

Reviewer #1: Yes

Reviewer #2: Yes

Reviewer #3: Yes

2. Has the statistical analysis been performed appropriately and rigorously? 

Reviewer #1: Yes

Reviewer #2: No

Reviewer #3: Yes

3. Have the authors made all data underlying the findings in their manuscript fully available?

Reviewer #1: Yes

Reviewer #2: Yes

Reviewer #3: Yes

4. Is the manuscript presented in an intelligible fashion and written in standard English?

Reviewer #1: Yes

Reviewer #2: Yes

Reviewer #3: Yes

5. Review Comments to the Author

Reviewer #1: This study evaluated the stability of seven reference gene candidates selected for RT-qPCR normalization in gene expression quantification, specifically for the four different sponge species. The work provided a panel of reference genes for RT-qPCR in species belonging to phylum of Porifera. It can be accepted for publication after the following questions are addressed and included in discussion as appropriate:

1. Only one species of sponges from each type (marine, freshwater, filter-feeding and carnivorous form) was used for validation, do you expect that other species of the same type will result in similar reference gene stability profile?

2. Many studies are conducted to examine gene expressions under specific stress conditions. How well do the results presented in this study translate to gene expression studies under conditions such as thermal stress, starvation, exposure to heavy metals, etc.?

3. How representative are the specimens from each species in representing the entire species?

4. For Figure 4c, 5, and S2, why are the data presented in dot plots with connected lines instead of bar graphs for presentation when reference gene stability ranking is not continuous? Please double check to see if the current data presentation is appropriate.

Reviewer #2: Line 22-33 Based on the experiment design and goals in M&M, the authors tested all candidate housekeeping genes from L. corallorrhiza on all four distinctive species to see if these housekeeping genes from L. corallorrhiza are universal for other three species. However, the abstract does not clearly verify this main goals, please reorganize Abstract.

Line 104-105 Please add this ‘hpo’ into the list of abbreviations used in manuscript

Line 105 and Line 116 Authors need to elaborate why sample size for IT (n=4), CR (24 hpo, n=6) and RM (48 hpo, n=4) is different.

Line 105-

Line 134 please give how much tissue in gram was used for RNA extraction per preparation per replicate

Line 134-137 and Line 139-142 repeated information, please adjust this. It is more suitable for subsection of ‘RNA extraction and reverse transcription”

Line 144-145 please add citation of RNA extraction method.

Line 144-162 please reorganize this section

Line 165-166 ‘b)’ were used for different transcriptome databases

Line 167-168 assemblies in Table S1 is not publicly available, it is upon the request, not sure if this is against PlosOne’s policy

Line 168 Please list the name and accession number of the query amino acid sequence used in search. What are criteria used in search ?

Line 174 ‘For lamins” ? please give more information of this in introduction section

Line 174-182 Please clarify this section. What is purpose of having this section here ? I believe you retrieved and used sequences from other species to search homologies targeted for housekeeping genes and validation gene from transcriptome databases of various sponge species, then rewrite this section.

Line 182 authors said the ‘accession IDs’ are available in Table S2 for NCBI and Uniprot, however the access ID for NCBI is empty.

Line 183-184 what is the size of amplicon for each target gene with designed primers ?

Line 199 Ta ? you mean Tm ?

Line 201-202 this sentence needs to be clarified

Line 204 if you use standard software with qPCR equipment, the software calculates the primer efficiency for you.

Line 232 genus name of each species could be abbreviated as H. , E. and L. as they were mentioned in previous sections

Line 242-243 Figure 1 does not match the results as said. Same to Figure 2 and Figure 3, double check the Figure legend.

Line 284 in Figure 4a panel, the letter assigned to each subpanel should be consistent with one in Figure legend, either in lower case or upper case. This should apply to other Figures too.

Line 298-327 all knowledge information of four analysis methods can be combined and move to introduction.

Line 310 For Figure 4d, please add values and conduct statistical comparisons. Same for Figure 6a.

Line 316-20 this part is more suitable for introduction.

Line 383-404 anova (statistical analysis) for multacomparisons are needed here to address and support your results.

Line 406 in discussion section, authors emphasized on the validation of housekeeping genes and their stability across species and tissue, this is one the key point of this manuscript. Meanwhile, the sequence comparisons and phylogenetic analysis prior to the candidate housekeeping gene selection is another key point authors brought into this manuscript, this makes this research unique and more valuable to scientists. Please elaborate this part more.

Line 494 In limitations section, authors listed a few potential issues when apply this protocol into other studies, which should be mentioned in abstract in a few words, such as ‘suitability of housekeeping genes should be reevaluated when …. before …”

For Figures

Please increase the font size to make them readable. Also add data at each data point in Figures.

Reviewer #3: Skorentseva et al used transcriptomic data sets to identify a suite of candidate reference genes based on similarity with known sequences and evaluated their suitability for poriferan RT-qPCR analyses. The manuscript is well-written, the methods employed are appropriate, and both the results and conclusions are supported by the data shown. While the study will undoubtedly prove to be a useful framework for future gene expression studies, the authors might consider more fully contextualizing how they envision their findings can be applied. In addition, it would be helpful if the authors expanded on their methodological details, particularly the immunostaining shown in Figure S1, which does not appear to be described in the Methods. More detailed comments are listed below.

Specific comments –

1) Methodological details.

- Provide accession numbers/gene names for the sequences used to query the reference databases.

- Define the criteria used to trim sequence reads with TrimAI.

- Were the qPCR products sequence validated?

- Provide methodological details for the immunostaining shown in Figure S1. Relatedly, the text on pg 9 references two antibodies for actin B, but only the polyclonal antibody is shown in Figure S1.

2) Pg 9, 2nd paragraph – Expression analysis of the candidate reference genes was evaluated across “all samples”. It would be helpful if the term “all samples” was more explicitly defined.

3) pg 10 – “TBP and LMN1 exhibited the highest variability”. However, Figure 4C shows that the lowest values were observed for LMN1 and HPRT1.

4) Clarify what sample sets/cDNAs were used to evaluate the candidate reference genes in the other species. Were the same time points/tissues as those referenced in Figure 4 used?

5) The results section describing data shown in Figure 6a indicates that the pattern of the NF primers relative to the individual primers was consistent across all experimental conditions. The IT1 sample set, however, does not show this pattern. Is this because the results were normalized to this dataset? If so, please state that in the Figure legend.

6) Unless it is a journal stipulation, the authors might consider moving the limitations section further up or incorporating the limitations directly into the Discussion.

7) Specifically label the y-axis in Figure 4B, 6B, and 6C.

Minor comments –

1) pg 2, 2nd to last line – check the spelling of Porifera

2) pg 3 - delete the spurious quote at the of the Introduction

3) pg 4 – define “hpo”; place a space between the integer value and the unit abbreviation (eg 0.5 mM rather than 0.5mM)

4) pg 5 – change “were soaked at” to “were dabbed on…”; change “0.1-10 ug of total RNA…” to “Total RNA (01-1 ug) was taken…”

5) pg 6 – change “Retrieved transcripts were…” to “The retrieved Porifera transcripts were…”

6) pg 7 – it is unclear if Ta is meant to be the annealing temp of the various primers, please define. Also, indicate that the primer specific temperatures are listed in Table S3.

7) pg 7 – change to “In this model, …”

8) pg 8 – change to “we aligned candidate transcripts to…cytoplasmic isoforms of vertebrate actin”

9) pg 9 – change to “To validate that the identified contigs encode cytoplasmic actins in sponges, L. corallhorrihiza regenerative membranes were immunostained with antibodies targeting various actins. Subsequent imaging revealed that cytoplasmic filamentous structures and stress fibers were stained with a certain level of differentiation between the different actin isoforms.” NOTE – the specific antibodies used can be moved to the methods section.

10) pg 11 – change to “…and RPS3A were among the most stably expressed…”

11) Figure 3 legend – something seems be missing in the parentheses

6. PLOS authors have the option to publish the peer review history of their article (what does this mean?). If published, this will include your full peer review and any attached files.

Reviewer #1: No

Reviewer #2: No

Reviewer #3: No

---

## [Author Response · Author response to Decision Letter 1]

31 Mar 2026

Manuscript ID: ONE-D-26-03346

Title: Validating the classics: accurate reference gene panel for reliable RT-qPCR in Porifera

PLOS One

Authors: Kseniia V. Skorentseva, Nikolai P. Melnikov, Alexander V. Ereskovsky, Andrey I. Lavrov & Alina A. Saidova

• A letter that responds to each point raised by the academic editor and reviewer(s). You should upload this letter as a separate file labeled 'Response to Reviewers'.

Journal Requirements:

The manuscript has been reformatted in accordance with the journal’s formatting guidelines and templates provided.

2. Thank you for stating the following financial disclosure: “The research was supported by the Russian Science Foundation project no. 23–74–10005 (studies in Leucosolenia corallorrhiza and Halisarca dujardinii) and no. 24-14-00452 (studies in Ephydatia fluviatilis and Lycopodina hypogea).”

If this statement is not correct you must amend it as needed. Please include this amended Role of Funder statement in your cover letter; we will change the online submission form on your behalf.

We have added the mentioned statement to the Funding section of the manuscript.

3. In the online submission form, you indicated that “A draft dataset supporting the conclusions of this study is available upon request. The R script used to analyze and visualize the data is also available.”

All PLOS journals now require all data underlying the findings described in their manuscript to be freely available to other researchers, either 1. In a public repository, 2. Within the manuscript itself, or 3. Uploaded as supplementary information. This policy applies to all data except where public deposition would breach compliance with the protocol approved by your research ethics board. If your data cannot be made publicly available for ethical or legal reasons (e.g., public availability would compromise patient privacy), please explain your reasons on resubmission and your exemption request will be escalated for approval.

In accordance with PLOS ONE’s data availability policy, we have deposited the raw dataset underlying the findings of this study, as well as the R script used for data analysis and visualization, in a public repository (Mendeley Data).

The materials are available at the following DOI: 10.17632/stkgvs9s26.1

The Data Availability Statement in the manuscript has been updated accordingly.

We confirm that the data underlying the findings of this study have now been made fully and freely available in a public repository (Mendeley Data).

The Data Availability Statement in both the manuscript and the online submission form has been updated accordingly to reflect that the data are publicly accessible at the provided DOI.

5. Please upload a new copy of Figure 2 as the detail is not clear. Please follow the link for more information: https://journals.plos.org/plosone/s/figures

We have uploaded a new version of Figure 2 in higher resolution to ensure that all details are clearly visible.

6. We notice that your supplementary [figures/tables] are included in the manuscript file. Please remove them and upload them with the file type 'Supporting Information'. Please ensure that each Supporting Information file has a legend listed in the manuscript after the references list.

We have removed all supplementary figures and tables from the main manuscript file and uploaded them separately under the file type “Supporting Information,” as requested. The files and manuscript have been updated accordingly.

We confirm that the reviewer comments did not contain any specific recommendations to cite previously published works. The suitable references have been added to the revised manuscript where appropriate.

Reviewers' comments:

Reviewer's Responses to Questions

Comments to the Author

1. Is the manuscript technically sound, and do the data support the conclusions?

Reviewer #1: Yes

Reviewer #2: Yes

Reviewer #3: Yes

2. Has the statistical analysis been performed appropriately and rigorously?

Reviewer #1: Yes

Reviewer #2: No

Reviewer #3: Yes

3. Have the authors made all data underlying the findings in their manuscript fully available?

Reviewer #1: Yes

Reviewer #2: Yes

Reviewer #3: Yes

4. Is the manuscript presented in an intelligible fashion and written in standard English?

Reviewer #1: Yes

Reviewer #2: Yes

Reviewer #3: Yes

5. Review Comments to the Author

Please use the space provided to explain your answers to the questions above. You may also include additional comments for the author, including concerns about dual publication, research ethics, or publication ethics. (Please upload your review as an attachment if it exceeds 20,000 characters).

We thank Reviewers for their positive comments. We appreciate the time they took to critically review the manuscript and to make recommendations that have significantly improved our manuscript. We have accepted majority of recommendations and hope that these revisions address the Reviewers’ criticisms.

Reviewer #1: This study evaluated the stability of seven reference gene candidates selected for RT-qPCR normalization in gene expression quantification, specifically for the four different sponge species. The work provided a panel of reference genes for RT-qPCR in species belonging to phylum of Porifera. It can be accepted for publication after the following questions are addressed and included in discussion as appropriate:

We sincerely thank the reviewer for their positive assessment and constructive comments. We have carefully considered all the reviewer’s suggestions and have addressed them in the revised manuscript. Relevant points have been incorporated into the Discussion section as appropriate.

1. Only one species of sponges from each type (marine, freshwater, filter-feeding and carnivorous form) was used for validation, do you expect that other species of the same type will result in similar reference gene stability profile?

We thank the reviewer for this insightful question.

As noted, the sponge species used in this study cannot be strictly classified into four distinct types. Instead, the four categories considered (marine, freshwater, filter-feeding, and carnivorous) represent combinations of these characteristics between different species: among the species used, there is one representative of calcareous sponges and three representatives of demosponges; additionally, there are one freshwater species, three marine species, three filter-feeding species, and a carnivorous one. Thus, each species combines these traits in different ways.

We agree that using a single representative species for each ecological or functional group (marine, freshwater, filter-feeding, and carnivorous) does not allow us to fully generalize the stability profiles to all species within these categories. However, our study was specifically designed to provide an initial cross-species validation across phylogenetically and ecologically diverse representatives of Porifera, rather than to establish universally stable reference genes for entire groups. Importantly, our results already indicate that no single gene is universally stable even across the four tested species, despite their grouping by ecological type. Instead, we observed that while the exact rankings vary, a consistent subset of genes (e.g., RPL13A, ACT1, GAPDH, and RPS3A) repeatedly appears among the most stable candidates across species. This suggests that, although species-specific validation remains necessary, there is a conserved core of relatively stable reference genes that can serve as a starting point for further studies. Based on these observations, we do not expect identical stability rankings in other species of the same ecological type. Rather, we expect that: (i) stability profiles will remain species-specific, and (ii) the same set of candidate genes is likely to perform robustly after validation, even if their ranking order differs. To emphasize this point, we have revised the Discussion section of the manuscript accordingly.

2. Many studies are conducted to examine gene expressions under specific stress conditions. How well do the results presented in this study translate to gene expression studies under conditions such as thermal stress, starvation, exposure to heavy metals, etc.?

We agree that reference gene stability can be influenced by experimental conditions such as thermal stress, starvation, or exposure to pollutants, and therefore results obtained under baseline conditions cannot be directly extrapolated to all stress contexts. In the present study, our goal was to identify candidate reference genes that are stable under standard physiological conditions across phylogenetically and ecologically diverse sponge species, thereby providing a baseline framework for normalization. Our results show that even under non-stressed conditions, there is no universally stable gene across species, highlighting the necessity of careful validation. Based on this, we expect that under stress conditions, gene expression variability may further increase, potentially altering the stability ranking of candidate reference genes. Therefore, we do not assume that the exact stability profiles identified here will remain unchanged under conditions such as thermal stress, starvation, or exposure to heavy metals. At the same time, we observed that several genes (e.g., RPL13A, ACT1, GAPDH, and RPS3A) consistently rank among the most stable candidates across multiple species. This suggests that these genes may represent a robust starting panel for reference gene selection, including in stress-related experiments. However, we emphasize that their suitability must be empirically validated under each specific experimental condition. We have now clarified this point in the Discussion, explicitly stating that environmental and physiological stressors may affect reference gene stability and that additional validation is required when applying these genes to stress-response studies and also added a corresponding statement to the Conclusion section of the manuscript to reflect this point.

3. How representative are the specimens from each species in representing the entire species?

All specimens used in the study were typical representatives of the corresponding species in sense of overall body structure, physiological condition (only specimens considered fully healthy was used in the study), habitat, etc. Culturing conditions for L. hypogea and gemmules of E. fluvatilis are well accepted and routinely used. The specimens of each species were indeed collected from relatively limited geographical areas, and we have added a description of this limitation to the ‘Limitations’ section of the manuscript. Nevertheless, we believe that this does not compromise the overall validity of our findings, as such a limitation is common in organismal biology studies conducted on non-model organisms.

4. For Figure 4c, 5, and S2, why are the data presented in dot plots with connected lines instead of bar graphs for presentation when reference gene stability ranking is not continuous? Please double check to see if the current data presentation is appropriate.

We have chosen to present the data in dot plots with connected lines following the approach used in a large body of previously published studies of this type [17, 39, 40, 46. We believe that this style of visualization facilitates comparison of data across different studies for future researchers.

Reviewer #2:

We sincerely thank Reviewer 2 for their thorough and careful evaluation of our manuscript. We greatly appreciate the detailed and constructive comments, which have helped us improve the clarity and quality of our work.

1. Line 22-33 Based on the experiment design and goals in M&M, the authors tested all candidate housekeeping genes from L. corallorrhiza on all four distinctive species to see if these housekeeping genes from L. corallorrhiza are universal for other three species. However, the abstract does not clearly verify these main goals, please reorganize Abstract.

We have revised the Abstract to clearly reflect the main goals of the study, including the testing of candidate housekeeping genes from L. corallorrhiza across the other three species, in accordance with the experimental design described in the Materials and Methods.

2. Line 104-105 Please add this

---

## [Decision Letter · Decision Letter 1]

30 Apr 2026

Validating the classics: accurate reference gene panel for reliable RT-qPCR in Porifera

PONE-D-26-03346R1

Dear Dr. Saidova,

We’re pleased to inform you that your manuscript has been judged scientifically suitable for publication and will be formally accepted for publication once it meets all outstanding technical requirements.

Kind regards,

Fang Zhu, Ph.D.

Academic Editor

PLOS One

Additional Editor Comments (optional):

Reviewers' comments:

Reviewer's Responses to Questions

**Comments to the Author**

1. If the authors have adequately addressed your comments raised in a previous round of review and you feel that this manuscript is now acceptable for publication, you may indicate that here to bypass the “Comments to the Author” section, enter your conflict of interest statement in the “Confidential to Editor” section, and submit your "Accept" recommendation.

Reviewer #1: All comments have been addressed

Reviewer #2: All comments have been addressed

Reviewer #3: All comments have been addressed

2. Is the manuscript technically sound, and do the data support the conclusions?

Reviewer #1: Yes

Reviewer #2: Yes

Reviewer #3: (No Response)

3. Has the statistical analysis been performed appropriately and rigorously? 

Reviewer #1: Yes

Reviewer #2: Yes

Reviewer #3: (No Response)

4. Have the authors made all data underlying the findings in their manuscript fully available?

Reviewer #1: Yes

Reviewer #2: Yes

Reviewer #3: (No Response)

5. Is the manuscript presented in an intelligible fashion and written in standard English?

Reviewer #1: Yes

Reviewer #2: Yes

Reviewer #3: (No Response)

6. Review Comments to the Author

Reviewer #1: (No Response)

Reviewer #2: Authors did great job on generating idea and collecting supporting data for this study. The written MS is readable and easy to follow for scientists. The data is also useful to other researchers in same area and expandable to other people in different biological area. Of one idea to select candidate housekeeping genes based on the copy number in transcriptome makes this paper unique.

Reviewer #3: (No Response)

7. PLOS authors have the option to publish the peer review history of their article (what does this mean?). If published, this will include your full peer review and any attached files.

Reviewer #1: No

Reviewer #2: No

Reviewer #3: No

---

## [Editor Report · Acceptance letter]

PONE-D-26-03346R1

PLOS One

Dear Dr. Saidova,

I'm pleased to inform you that your manuscript has been deemed suitable for publication in PLOS One. Congratulations! Your manuscript is now being handed over to our production team.

Kind regards,

on behalf of

Dr. Fang Zhu

Academic Editor

PLOS One